# Explainable Multi-Objective Model Selection for Time Series Forecasting

## Abstract

Machine learning (ML) models exhibit miscellaneous properties, and deployment inevitably trades certain performance aspects against others. This is particularly valid for time series forecasting, where special characteristics such as seasonality impact how models perform. Since there is "no free lunch", practitioners have to choose among available methods when assembling new learning systems. Benchmarks, meta-learning, and automated ML come to aid, but in many cases focus on predictive capabilities while ignoring other aspects such as complexity and resource consumption. This is especially concerning considering the popularity of deep neural networks (DNNs) for forecasting, as these models are widely conceived as computation-heavy black boxes. To alleviate these shortcomings, we propose X-PCR – a novel approach for explainable multi-objective model selection. It uses meta-learning to assess the suitability of any model in terms of (p)redictive error, (c)omplexity and (r)esource demand. By allowing users to prioritize the individual objectives in this trade-off, model recommendations become both controllable and understandable. We demonstrate the feasibility of our methodology in the task of forecasting time series with state-of-the-art DNNs. In total, we perform over 1000 experiments across 114 data sets, discuss the resulting efficiency landscape, and provide evidence of how X-PCR outperforms other selection approaches. On average, our approach only requires 20% of computation costs for recommending models with 85% of the best possible performance.

## 1 Introduction

The rapidly evolving field of machine learning (ML) has brought forth a broad arsenal of learning methods, each exhibiting specific properties related to *predictive* performance, model *complexity* and computational *resource* demand. Deployment requires evaluating the pros and cons of applicable methods, and as there is "no free lunch", these trade-offs vary vastly across data sets. This is the case for almost every learning task, including time series forecasting, which due to the complex and time-evolving nature of this data, has always been considered a very challenging task (Saadallah et al., 2019; 2022). The relevance of model properties usually depends on the particular application at hand. For instance, in safety-critical scenarios, robustness (Croce et al., 2020) and predictive precision are crucial, whereas for edge devices (Buschjäger et al., 2020), model size and computational resource demand are tightly constrained. Explicitly investigating these properties is especially important when considering today's popularity of deep neural networks (DNNs) (Saadallah et al., 2021). While they have been successfully utilized for time series forecasting (Livieris et al., 2020; Kim & Cho, 2019), their complex structure also makes them computationally expensive (Strubell et al., 2020) and non-transparent – sometimes resulting in them being referred to as black boxes (Molnar, 2020).

Selecting the "best" model for a given learning task can be understood as a *multi-objective optimization problem*, with individual objectives representing model properties as prioritized by the use case at hand. Even with limiting the search to DNNs, the space of forecasting models is huge (Alexandrov et al., 2020). Practically, this renders the identification of optimal models via exhaustive search (i.e., testing all options) computationally redundant and, quite possibly, unfeasible. Benchmarks such as the Monash forecasting repository by Godahewa et al. (2021) are helpful for model selection. However, they are usually limited to reporting predictive performance. Automated ML (AutoML) pipelines (Jin et al., 2019) suffer from the same phenomenon and, moreover, do not consider the

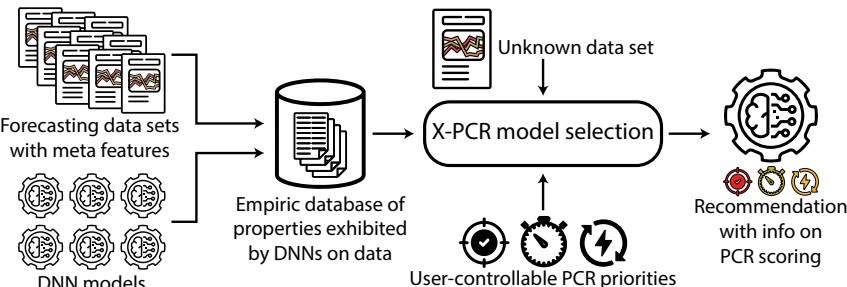

Figure 1: Framework for X-PCR selection. Our method leverages meta-learning to estimate DNN performance on a given data set in terms of prediction error, complexity and resource demand. The final recommendation aims at finding the best trade-off considering all estimated properties.

unique characteristics of time series forecasting (Alsharef et al., 2022). Even though the necessity of resource-awareness has been highlighted both generally (Fischer et al., 2022) and for time series (Uchiteleva et al., 2021), there is no consistent guideline for ensuring an efficient quality-resource trade-off during model selection. Another major shortage of existing frameworks is the lack of transparency. AutoML specifically aims at automating the process of model selection and hyperparameter tuning, making it difficult to understand the reasons behind model recommendations. Moreover, aspects of model explainability related to network complexity are often neglected during the search. This is highly contrary to the widely supported call for trustworthy ML (Brundage et al., 2020; EU AI HLEG, 2020), which should also be considered when building forecasting applications.

Our work addresses these challenges by proposing **X-PCR** model selection and demonstrating its feasibility for time series forecasting with DNNs. The approach leverages multi-objective meta-learning, considering time series' characteristics as features and recommending models based on their suitability in terms of (p)rediction error, (c)omplexity and (r)esource consumption. As expected with multi-objective problems, there could actually be multiple Pareto-optimal solutions for any model search. Note, however, that classic optimization is inapplicable for this task, since probing the search space (i.e., testing single model options) is highly expensive. Our approach reduces this effort by estimating any model option's multi-objective performance across PCR properties based on the available meta-features. In addition, we claim our proposed method to be e(x)plainable on three levels: Firstly, the selection process itself becomes explainable by making scoring along all PCR objectives available with any recommendation. Secondly, by enabling users to align the PCR criteria with their priorities, the process becomes highly interactive and, thus, more transparent. Lastly, we use per-construction interpretable meta-modeling (Rudin, 2019) to further explain the reasoning behind any recommendation. Our framework is schematically displayed at a glance in Figure 1.

As part of our work, we also provide an extensive experimental evaluation showcasing both the implicit efficiency trade-offs when forecasting time series with DNNs, as well as the capabilities of X-PCR selection. 11 state-of-the-art (SOTA) DNNs were deployed on over 100 data sets, resulting in a total of 1254 experiments for our investigations – the complete implementation and logs are available at `github.com/tmplxz/xpcr` (preliminarily published for reviewers). The results are evidence of how considering PCR properties completely changes the understanding of SOTA in forecasting. Our selection method beats competing approaches such as AutoML and achieves 85% of predictive performance at only 20% of the computation cost required for exhaustive search. In nearly all cases, the optimal model is found within the top-5 recommendations. The evaluation shows that our novel X-PCR selection is a beneficial contribution to SOTA research on resource-aware ML, explainable ML, and time series forecasting.

## 2 RELATED WORK

Before explaining the intricacies of our approach, we present a review of forecasting with DNNs, a literary background on model selection and meta-learning, as well as recent works investigating the importance of explainability and resource-awareness in ML.

## 2.1 On DNNs for Time Series Forecasting

Deep learning has been successfully applied to forecasting (Livieris et al., 2020; Kim & Cho, 2019). At present, the domain is mostly dominated by recurrent (RNNs), particularly long short-term memory (LSTM), neural networks. Their major benefit lies in the capability of learning from the entire history of series values (Kim & Cho, 2019). Convolutional networks (CNNs) also score well in forecasting and tend to be more efficient than RNNs, as convolutions allow for expanding the receptive field exponentially while maintaining a small parameter space (Borovykh et al., 2017). The recently published GluonTS toolkit (Alexandrov et al., 2020) offers a diverse range of recent DNN models for usage on time series, discussed extensively in a review by Benidis et al. (2020). DeepAR employs a RNN with LSTM or Gated Recurrent Unit cells (Salinas et al., 2020), while DeepState learns RNN weights jointly across all time series via a Kalman filter (Rangapuram et al., 2018). MQ-RNN and MQ-CNN leverage RNN and dilated causal CNN encoders, respectively, coupled with a quantile decoder, establishing sequence-to-sequence prediction (Wen et al., 2017). DeepFactor also estimates weights across series, combining a local approach with a global DNN (Wang et al., 2019). With Transformers (Lim et al., 2021) and Wavenet (Oord et al., 2016), GluonTS also offers architectures popular in natural language processing. Unsurprisingly, no model is universally valid for every forecasting application (Godahewa et al., 2021; Saadallah et al., 2019; Cerqueira et al., 2019).

## 2.2 Model Selection

Various techniques have been proposed for selecting an appropriate model from a set of candidates for a specific task. Firstly, one can try to estimate each candidate's expected error, e.g. via Gaussian (Birgé & Massart, 2001) or Bayesian estimation methods. These methods are however impractical for forecasting, since they require to approximate continuous composite densities for the error between target and estimated time series values. Each candidate's error can also be estimated based on empirical evaluations (Rivals & Personnaz, 1999), such as the forecasting benchmark contributed by Godahewa et al. (2021). Naturally, this approach is always limited to the empirically investigated data, methods and properties. Implementing a meta-learning paradigm allows to learn additional ML models that predict the behaviour based on previous selections characterized by a set of meta-features (Wolpert, 1992). Meta-learning has been successfully used in forecasting (Cerqueira et al., 2017; Saadallah et al., 2019), for example by specializing a set of candidate models over different series regions (Saadallah et al., 2022). AutoML can be understood as meta-learning without explicitly formulating meta-features While being popular in deep learning (Jin et al., 2019; Zimmer et al., 2021), it currently does not handle sequential dependencies, variable-length inputs, and dynamic temporal patterns specific to forecasting well. In addition to the already mentioned drawbacks, the established approaches for model selection suffer from exclusively focusing on predictive performance. As such, they do not consider other significant model-allied properties such as complexity or resource consumption.

## 2.3 Explainability and Resource-aware ML

With the race for bigger models, many works have highlighted the importance of trustworthiness (Brundage et al., 2020), which eventually manifested in the EU AI act (EU AI HLEG, 2020). The demand for more trust sparked research on explainability (Samek et al., 2019) that makes complex models like DNNs more interpretable (Rudin, 2019). With ML becoming a popular tool across many domains, works have also motivated to specifically bridge the communication gap towards non-experts (Morik et al., 2022). This includes aspects of sustainability and resource-awareness (Patterson et al., 2021), as modern ML was shown to significantly impact our environment (Strubell et al., 2020). As a result, recent works investigated carbon emissions in areas such as computer vision (Schwartz et al., 2020), data stream mining (García-Martín et al., 2019), and language models (Patterson et al., 2021; Bender et al., 2021; Strubell et al., 2020). In order to make the SOTA more resource-aware, the ML community is required to establish practices for reducing ML carbon emissions (Patterson et al., 2022; Lacoste et al., 2019) and improve our reporting in terms of resource efficiency (Fischer et al., 2022). On the contrary, model selection approaches usually base their recommendation only on predictive performance, and other properties than predictive performance are rarely reported (Godahewa et al., 2021). Our work aims at increasing both the understanding and resource efficiency of forecasting models by explicitly embedding these aspects into our novel selection approach.

## 3 EXPLAINABLE MULTI-OBJECTIVE MODEL SELECTION

Our methodology understands any ML experiment to be characterized by an underlying configuration and environment. As established by Fischer et al. (2022), the former specifies model, data set, and task, while the latter represents the software and hardware platform for practical execution While we specifically formulate our method for the task of *learning and evaluating a DNN on forecasting data*, it can be easily generalized to other ML domains. Our framework scheme in Figure 1 functions as a guide through our methodology.

### 3.1 PROBLEM STATEMENT

Let $Y$ be a time series, i.e., a temporal sequence of values, where $Y_{1:t} = \{y_1, y_2, \cdots, y_t\}$ is a sequence of $Y$ recorded until time $t$ and $y_i$ is the value of $Y$ at time $i$. In most applications, $Y$ is not recorded in isolation but in a wider context enclosing many time series variables $Y \in \mathbf{Y}$. As an example consider weather data, where single time series $Y$ might represent temperature, wind speed, or humidity recordings, which together form $\mathbf{Y}$. Let $M$ be a pool of DNNs that can be applied to $\mathbf{Y}$ by performing univariate forecasting for each single series. Selecting the best DNN can then be formally understood as a multi-objective optimization problem (Yang, 2014):

$$\operatorname*{argmax}_{m \in M}(F(\mathbf{Y}, m)) = \operatorname*{argmax}_{m \in M}(\sum_{i=1}^{k} w_i f_i(\mathbf{Y}, m)) \text{ with } \forall i, w_i \geq 0 \text{ and } \sum_{i=1}^{k} w_i = 1 \qquad (1)$$

The *compound score* $F(\mathbf{Y}, m)$ is a weighted sum of functions $f_i$, which describe empirical properties that $m$ exhibits when being applied to $\mathbf{Y}$. As these can be grouped into describing *(p)rediction* error, *(c)omplexity*, or *(r)esource* consumption, we name the $f_i$ *PCR functions*. To give some examples, one could assess the prediction error on unseen test data (P), number of model parameters (C), or power draw for a single inference step (R). Following Equation (1), the individual PCR functions should be defined such that maximization leads to improved model behaviour. The weights $w_i$ allow for prioritizing certain properties depending on the use case at hand. This is of high relevance for explicitly making models more resource-aware or explainable, since these aspects are directly linked to any model's resource demand and complexity. Note that solving Equation (1) is non-trivial: Firstly, the functions $f_i$ can behave contradictorily, which complicates simultaneous optimization. To give an example, prioritizing less complex models usually results in higher prediction errors. Also, depending on the problem and model space at hand, there might not even be a single solution to Equation (1), but rather a range of so-called Pareto-optimal choices. We want to explicitly stress that probing this front is computationally expensive, as it requires to train and evaluate models in $M$. Moreover, it is impossible to calculate derivatives of the PCR functions $f_i$, since they are usually assessed from empiric behavior of each respective model. As a result, classic optimization approaches cannot be applied to search the multi-dimensional space for Pareto points.

### 3.2 COMPARABILITY OF PROPERTIES

Assessing the function values for $f_i$ unveils a major problem: The numeric behavior and meaning of properties will be vastly different – we could either obtain hundreds of milliseconds for running time, dozens of kilowatt-hours for power draw, or millions of parameters. This problem gets even more evident when considering different environments for running models (e.g., CPU or GPU implementations), as they will dramatically change the value magnitude. We address this issue by calculating values of $f_i$ on a relative *index scale* as proposed by Fischer et al. (2022). It is based on the real measurements $\mu_i$, which are obtained from applying $m$ to $\mathbf{Y}$. Whereas the original work proposed to calculate index values based on reference models, we instead resort to using the empirically best-performing model $m^*$ on the $i$-th property:

$$f_i(\mathbf{Y}, m) = \left( \frac{\mu_i^*(\mathbf{Y})}{\mu_i(\mathbf{Y}, m)} \right) = \left( \frac{\min_{m^* \in M}(\mu_i(\mathbf{Y}, m^*))}{\mu_i(\mathbf{Y}, m)} \right), \qquad (2)$$

The now calculable PCR function values $f_i$ and compound score $F$ are bounded by the interval $(0, 1]$ and describe the behavior in the given environment relatively; the higher the value, the closer it is to the best empiric result which receives $f_i(\mathbf{Y}, m) = 1$. Note that using the best value per property as reference is advantageous to global reference models since it neatly solves the problem of choosing reference models.

### 3.3 Model Selection using Meta-learning

Naively, Equation (1) can be solved by performing an *exhaustive search* that determines the PCR properties of all DNNs in $M$. While guaranteed to provide optimal solutions, this also requires dramatic computational expenses. Instead, we propose to adapt meta-learning and estimate the expected values of $f_i$ given the characteristics of input data $\mathbf{Y}$ and model option $m$. Accordingly, the meta-task can be defined as finding regression models $\hat{f}_i \in \mathcal{M}$, which are trained to model the property functions $f_i$. The space of possible models $\mathcal{M} : \mathcal{F}_{\mathbf{Y}} \times \mathcal{F}_m \rightarrow \mathbb{R}$ is defined by the space of model ($\mathcal{F}_m \subset \mathbb{R}^{n'}$) and data ($\mathcal{F}_{\mathbf{Y}} \subset \mathbb{R}^n$) meta-features. To be more specific, features $X_m \in \mathcal{F}_m$ encode general information about the model $m$, like type of network layers, while features $X_{\mathbf{Y}} \in \mathcal{F}_{\mathbf{Y}}$ encode information like seasonality or stationarity that describe the temporal data. More details about the meta-features in our experiments are given in Section 4.2. Training the meta-regressors $\hat{f}_i$ requires collecting a *property database* $D$ of meta-features and PCR function values for different configurations, i.e. , $D = \{(X_{\mathbf{Y}}, X_m; f_i(\mathbf{Y}, m))\}$. With the aforementioned "no free lunch" theorem in mind, cross-validatation across the database helps identify the best regression method for each property, i.e., the one with lowest reconstruction error:

$$\min_{\hat{f}_i} \sum_{(X_{\mathbf{Y}}, X_m; f_i(\mathbf{Y}, m)) \in D} |f_i(\mathbf{Y}, m) - \hat{f}_i(X_{\mathbf{Y}}, X_m)| \tag{3}$$

Given the property weights and meta-features of a new data set, one can now estimate solution(s) for Equation (1) by running regression queries for the meta-feature of all models, instead of redundantly applying them to $\mathbf{Y}$. By replacing all $f_i$ for the meta-predictions $\hat{f}_i$, we can easily calculate estimated compound scores $\hat{F}(x)$. To give an alternative approach, one could train a global recommendation model to *directly* learn the compound scores $F$. However, this would not allow for estimating or controlling the individual PCR properties, which makes the model selection less transparent and interactive. We later also show empiric evidence that this approach - as expected - does not perform better than a compositional recommender, which uses Equation (1) to aggregate the outputs of individually trained models.

### 3.4 Explainability Aspects

We argue our presented PCR-aware model selection to be explainable as it offers insights on many different levels. Firstly, for any query, our method provides by-product explanations in the form of estimates for all property functions. They inform interested users about the recommendation's estimated PCR trade-offs, explaining to what extent it is expected to exhibit each property. Thanks to the relative index scales (recall Section 3.2), this information highly comprehensible in itself: a score of $0.4$ implies that this model achieves $40\%$ of the best empiric result observed in $D$. Our model selection can be made even more explainable by explicitly using interpretable meta-regressors (Rudin, 2019). More precisely, meta-regressors equipped with probabilities or feature importance enable to investigate the link between model recommendations and certain data characteristics. Performing a sensitivity analysis would allow to go even deeper, and explore how changing these characteristics impacts the model recommendation.

In addition, interactions have been shown to potentially improve trust and explainability (Beckh et al., 2023), which also applies to our framework. By interactively prioritizing individual properties, the selection process becomes controllable and thus transparent. It also enables users to specifically prioritize less complex, or in other words, more interpretable models. Lastly, model recommendations and PCR trade-offs can be made more comprehensible to non-experts via discrete ratings and informative labels (Fischer et al., 2022) - this is readily implemented in our exploration tool. In summary, all these factors contribute to trusting our proposed model selection.

## 4 Experiments

We now investigate the practicability of our methodology. To assemble the 114 data sets, we split 19 original Monash data sets (Godahewa et al., 2021) from various domains into five cross-validation groups and increased their size by sub-sampling variants of each data set. We tested all DNNs in the benchmark, namely Feed-Forward (FFO), DeepAR (DAR), N-BEATS (NBE) and WaveNet

Table 1: Forecasting properties with impact on compound score

| Property | Group | Weight |
|---|---|---|
| Test MASE | Performance | 0.111 |
| Test RMSE | Performance | 0.111 |
| Test MAPE | Performance | 0.111 |
| Number of Parameters | Complexity | 0.167 |
| Model Size on Disc | Complexity | 0.167 |
| Training Power Draw | Resources | 0.083 |
| Training Time | Resources | 0.083 |
| Power Draw per Inference | Resources | 0.083 |
| Running Time per Inference | Resources | 0.083 |

Table 2: Error measures for assessing the meta-learner quality

| | |
|---|---|
| (a) | $\epsilon = \|f - \hat{f}\|$ |
| (b) | $\epsilon \overset{!}{<} 0.1$ |
| (c) | $\max_m f \overset{!}{=} \max_m \hat{f}$ |
| (d) | $\max_m f \in \{\text{top-}5_m\, \hat{f}\}$ |
| (e) | $\{\text{top-}5_m\, f\} \cap \{\text{top-}5_m\, \hat{f}\}$ |

Table 3: PCR-aware performance (compound score) of DNNs across Monash data sets

| Data set | DAR | DFA | DRP | DST | FFO | GPF | MQC | MQR | NBE | TFT | WVN |
|---|---|---|---|---|---|---|---|---|---|---|---|
| Austr..and | 0.21 | 0.05 | 0.11 | 0.18 | **0.73** | 0.34 | 0.12 | 0.06 | 0.32 | 0.28 | 0.22 |
| Car Parts | 0.36 | **0.64** | 0.52 | 0.27 | 0.63 | 0.58 | 0.59 | 0.57 | 0.21 | 0.37 | 0.27 |
| CIF 2016 | 0.26 | 0.22 | 0.22 | 0.15 | **0.72** | 0.52 | 0.48 | 0.25 | 0.22 | 0.28 | 0.16 |
| Dominick | 0.43 | 0.43 | 0.45 | 0.34 | **0.75** | 0.48 | 0.52 | 0.51 | 0.23 | 0.63 | 0.33 |
| Elect..kly | 0.40 | 0.32 | 0.23 | 0.27 | **0.73** | 0.41 | 0.46 | 0.18 | 0.24 | 0.35 | 0.19 |
| FRED-MD | 0.24 | 0.40 | 0.24 | 0.14 | **0.71** | 0.55 | 0.40 | 0.20 | 0.21 | 0.33 | 0.22 |
| Hospital | 0.43 | 0.39 | 0.26 | 0.44 | **0.69** | 0.59 | 0.61 | 0.42 | 0.24 | 0.45 | 0.29 |
| M1 Monthly | 0.38 | 0.44 | 0.26 | 0.40 | 0.61 | **0.69** | 0.51 | 0.24 | 0.21 | 0.42 | 0.16 |
| M1 Qu..rly | 0.36 | 0.40 | 0.30 | 0.35 | **0.68** | 0.59 | 0.53 | 0.30 | 0.26 | 0.32 | 0.31 |
| M3 Monthly | 0.37 | 0.53 | 0.25 | 0.35 | **0.60** | 0.53 | 0.57 | 0.40 | 0.28 | 0.38 | 0.29 |
| M3 Qu..rly | 0.35 | 0.40 | 0.26 | 0.41 | **0.62** | 0.44 | 0.49 | 0.28 | 0.28 | 0.33 | 0.33 |
| M4 Hourly | 0.22 | 0.33 | 0.25 | 0.32 | **0.62** | 0.22 | 0.21 | 0.06 | 0.19 | 0.20 | 0.25 |
| M4 Weekly | 0.38 | 0.33 | 0.25 | 0.30 | **0.71** | 0.44 | 0.69 | 0.27 | 0.27 | 0.43 | 0.27 |
| NN5 Daily | 0.36 | 0.59 | 0.32 | 0.35 | **0.64** | 0.47 | 0.57 | 0.47 | 0.23 | 0.29 | 0.33 |
| NN5 Weekly | 0.38 | 0.37 | 0.23 | 0.33 | **0.71** | 0.60 | 0.50 | 0.44 | 0.26 | 0.32 | 0.25 |
| Solar..kly | 0.43 | 0.38 | 0.28 | 0.07 | **0.65** | 0.60 | 0.59 | 0.33 | 0.20 | 0.30 | 0.15 |
| Touri..hly | 0.34 | 0.37 | 0.29 | 0.28 | 0.59 | **0.62** | 0.46 | 0.27 | 0.26 | 0.27 | 0.17 |
| Touri..rly | 0.47 | 0.38 | 0.29 | 0.28 | **0.72** | 0.47 | 0.52 | 0.24 | 0.25 | 0.32 | 0.24 |
| Traff..kly | 0.34 | 0.43 | 0.35 | 0.26 | **0.72** | 0.46 | 0.47 | 0.50 | 0.24 | 0.33 | 0.22 |

(WVN), and extended it to also include DeepFactor (DFA), DeepState (DST), Deep Renewal Processes (DRP), GPForecaster (GPF), MQ-CNN & MQ-RNN (MQC & MQC) and Temporal Fusion Transformer (TFT), totaling in $|M| = 11$ competing models (literary provided in Section 2.1). Table 1 lists the nine properties for functions $f_i$ in Equation (1), which describe training and inference behaviour. Weights $w_i$ were chosen to mitigate correlations (e.g., different errors) but still maintain a sound trade-off – in sum, each PCR group is equally weighted. Besides the well-known root mean squared error (RMSE), we also investigate the mean absolute scaled (MASE) and percentage (MAPE) error that are specialized for time series (Hyndman & Koehler, 2006). Our implementation uses GluonTS (Alexandrov et al., 2020) for deep learning, CodeCarbon (Schmidt et al., 2021) for profiling and Scikit-learn (Pedregosa et al., 2011) for meta-learning. All experiments were performed on a single PC with Xeon W-2155 CPU and took about two weeks, with total estimated carbon emissions of 24 $CO_2$e (Lacoste et al., 2019). Note that the computational effort stems mostly from training the $114 \times 11 = 1254$ DNNs – running X-PCR selection (i.e., training and evaluating the meta-regressors on the property database) only requires a few seconds. Code, hyperparameters, and logs are available at github.com/tmplxz/xpcr, including an exploration tool for interactively discovering the forecasting efficiency landscape (preliminarily published for reviewers).

## 4.1 PCR TRADE-OFFS WHEN FORECASTING WITH DNNs

Firstly, we improve upon the benchmark of Godahewa et al. (2021) by investigating the PCR trade-off instead of merely reporting error measures. Table 3 lists the compound scores obtained from Equation (1) for each Monash data × DNN combination. It clearly shows how each method's capabilities change with the data at hand, but generally, FFO appears to make good efficiency trades. To investigate this in more detail, we can explore the multi-dimensional space of properties. Exemplary, we display the inference running time versus MASE and training power draw versus RMSE

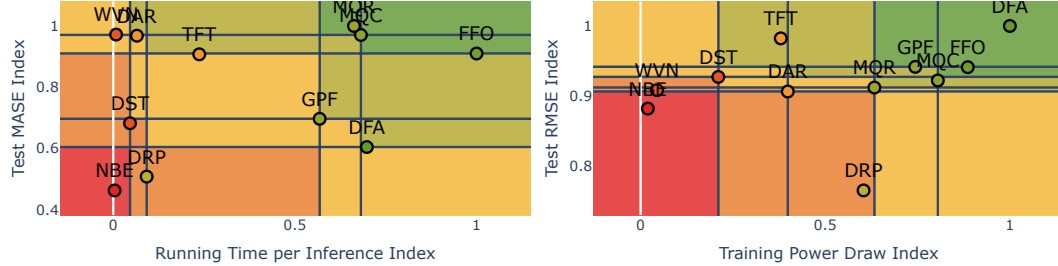

Figure 2: Trade-off between inference running time, training power draw, and prediction errors on Car Parts data (maximization indicating better performance). Point colors express the compound score, with four highly influential factors being displayed here. DST and FFO seem to trade best.

Table 4: Prediction error [MASE] and power draw [kWh] of different model search approaches

| Data set | X-PCR | | Random | | AutoKeras | | Exhaustive | |
|---|---|---|---|---|---|---|---|---|
| | MASE | kWh | MASE | kWh | MASE | kWh | MASE | kWh |
| Austr..and | 1.687 | **29.742** | **1.257** | 86.945 | 13.029 | 69.940 | 1.005 | 913.82 |
| Car Parts | **0.471** | **1.554** | 0.668 | 2.943 | 0.971 | 8.471 | 0.456 | 56.392 |
| CIF 2016 | **1.029** | **0.327** | 1.091 | 3.528 | 16.289 | 2.456 | 1.029 | 9.862 |
| Dominick | 1.774 | **28.917** | **1.580** | 30.666 | 2.535 | 245.08 | 1.469 | 1000.4 |
| Elect..kly | **2.793** | **0.971** | 17.452 | 2.213 | 19.886 | 12.232 | 1.653 | 52.556 |
| FRED-MD | **1.923** | 0.566 | 32758. | **0.160** | 990.52 | 6.321 | 0.655 | 12.307 |
| Hospital | **0.761** | **3.923** | 0.811 | 15.572 | 75.563 | 20.029 | 0.761 | 52.238 |
| M1 Monthly | **1.498** | **4.094** | 1.744 | 12.836 | 1171.8 | 21.620 | 1.359 | 55.363 |
| M1 Qu..rly | **1.783** | 0.855 | 2.329 | **0.415** | 334.97 | 2.228 | 1.783 | 16.553 |
| M3 Monthly | 1.164 | 11.098 | **1.077** | **5.226** | 2.573 | 80.654 | 1.018 | 272.87 |
| M3 Qu..rly | 2.085 | 3.202 | 120734 | **0.493** | 2.673 | 7.807 | 1.321 | 86.256 |
| M4 Hourly | **3.137** | 7.155 | 9.423 | **5.145** | 77216. | 212.89 | 2.801 | 414.00 |
| M4 Weekly | 4.134 | 4.519 | 71.787 | **1.593** | 25.706 | 24.626 | 2.743 | 71.024 |
| NN5 Daily | **0.684** | 0.802 | 1.497 | **0.192** | 1.212 | 29.775 | 0.568 | 42.706 |
| NN5 Weekly | **0.880** | 1.561 | 0.949 | **0.194** | 1.578 | 2.164 | 0.839 | 38.942 |
| Solar..kly | **0.829** | **0.483** | 1.049 | 0.867 | 1.004 | 1.242 | 0.829 | 23.939 |
| Touri..hly | **1.435** | 3.296 | 5.711 | **1.446** | 6.941 | 41.297 | 1.435 | 58.523 |
| Touri..rly | **1.670** | **0.662** | 5.741 | 3.034 | 12.942 | 11.994 | 1.670 | 24.203 |
| Traff..kly | **1.534** | **5.720** | 3.204 | 12.969 | 5.060 | 9.744 | 1.275 | 106.78 |

comparisons for the Car Parts data in Figure 2. Each scatter point represents a possible DNN option colored according to its compound score (also given in Table 3, with four out of nine contributing properties shown as axes. Keep in mind that due to the index scaling introduced in Section 3.2, each value needs to be maximized, with 1 indicating the best possible result. For this particular data set, DFA and FFO score best across all properties.

## 4.2 CAPABILITIES OF X-PCR MODEL SELECTION

For testing X-PCR, we devised meta-features that describe the model and data. They constitute information on seasonality and forecast horizon of each data set (given in the Monash code repository), averaged statistics across all series (length, mean, min, max), as well as one-hot encoded information on model choice, totaling in a data shape of $(1254, 21)$. Regarding meta-learner options, we restricted the search space $\mathcal{M}$ to simple and interpretable models (Rudin, 2019) like support vector regressors and decision trees - exact details are given in our code base. The best regressor per property was determined via the aforementioned grouped cross-validation, which in order to prevent information leak assigned all variants of a data set to either train or test split.

Let us first investigate the efficiency of utilizing X-PCR selection over randomly choosing a model per data set, deploying AutoKeras (Jin et al., 2019), or exhaustively testing all suggested DNNs. The resulting MASE and computational effort of each model search approach is given in Table 4. It clearly shows the superiority of X-PCR selection, which in most cases beats random selection and performs dramatically better than AutoKeras. Solving the problem via brute force by exhaustively

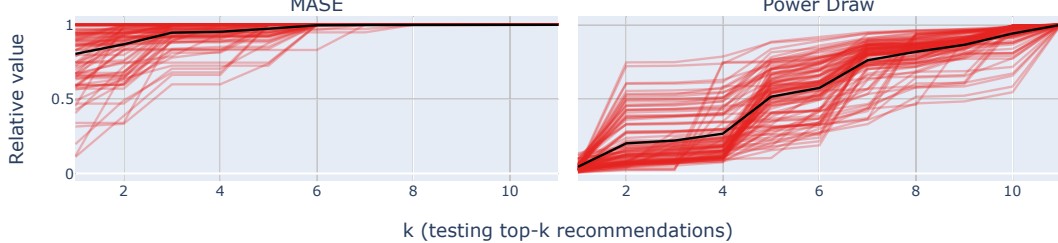

Figure 3: Relative best MASE and required power draw of testing the top-$k$ recommended models for all 114 data sets (red) and averaged (black). The relative values are measured based on the naive approach of testing all models for finding the optimum. Testing the top two models provides an average relative MASE of 85% at less than 20% of the energy cost.

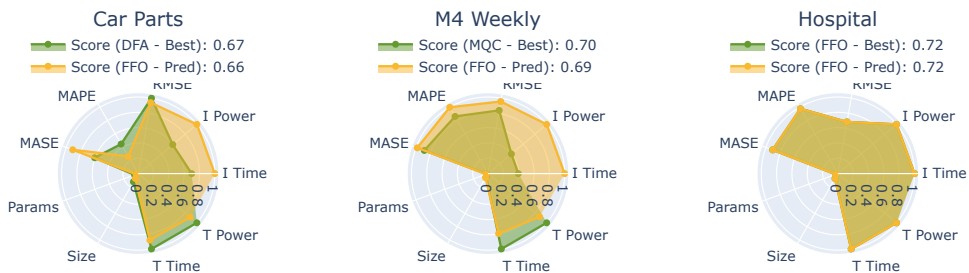

Figure 4: Properties of optimal and best-recommended models. Individual properties can diverge, but the compound scores are very close. On Hospital data, the actual best model was recommended.

training all models sometimes achieves even lower errors, however, it also requires much more computational effort. By testing the top-$k$ recommended models, we can evaluate how many X-PCRrecommendations are needed to get optimal results. This is visualized for all 114 data sets in Figure 3, with MASE and power draw being compared relatively to exhaustively testing all DNNs. On average (black), 85% of the best possible MASE can already be achieved by just testing the two models scored best by X-PCR, requiring less than 20% of the amount of energy. In nearly all cases, the best-performing model will be under the top-5 recommendations.

By now it should have become clear that any model's performance on particular data needs to be understood as a point in a complex space of properties. We exemplary visualize these multi-dimensional trade-offs of actual best and recommended DNNs via star plots in Figure 4. We purposefully selected two data sets where X-PCRfails to recommend the best model to obtain diverging star shapes. It shows how both models behave quite differently, but still score well on Equation (1).

To demonstrate the explainability of X-PCRmodel selection, Figure 5 shows exemplary explanations of our method. It firstly informs on which (estimated) properties mostly contribute to the estimated compound score, or in other words, support the model recommendation. In this example, we see that the estimated MASE of FFO makes it a favorable choice for the Car Parts data. Our method also allows users to dig deeper by querying the interpretable property estimation models for meta-feature importance. In this case, we see that besides model choice, the number of series and average series minimum value mostly affected the MASE estimate.

Lastly, we investigate the difference between real function values $f_i$ and predicted $\hat{f}_i$, as well as resulting compound scores $F$ and $\hat{F}$ (recall Equation (3)). For that, we assess the prediction error (a), the accuracy of scoring an error below a threshold of 0.1 (b), top-1 (c) & top-5 (d) accuracy of predicting the best obtainable result, and the intersection size of top-5 recommendation and top-5 true best models (e), as defined in Table 2. These error measures are provided in Figure 6 for each individual property, as well as the compositional and directly estimated compound scores as explained in Section 3.3. As expected, some properties like training time are harder to predict, while others (e.g., number of parameters) seem to behave rather deterministic. The compositional recommender approach outperformes directly estimating the compound scores, which has slightly higher errors.

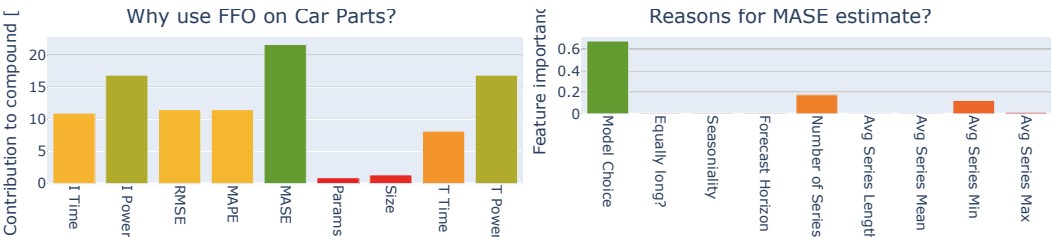

Figure 5: Explanations for model recommendation (property contribution to the compound score estimate – left) and feature importance for any property assessment (here shown for MASE – right).

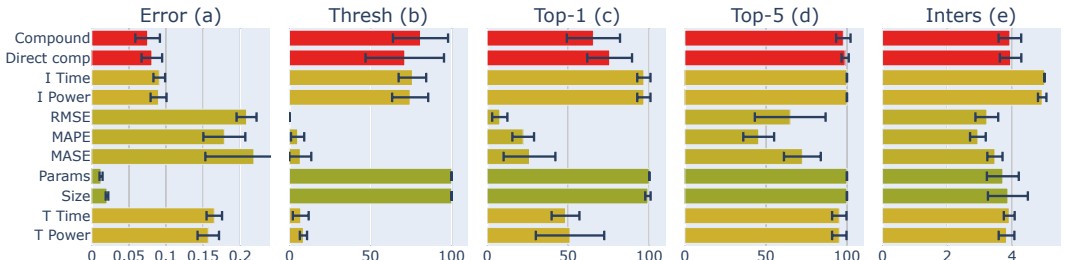

Figure 6: Quality of estimating different properties and compound score. Some properties (e.g., training time) are harder to predict than others, resulting in higher errors. Directly estimating compound scores performs worse than using our proposed compositional approach. The top-5 accuracy and intersection are evidence of how testing several recommendations has a high chance of retrieving the optimal solution. Colors indicate each property's contribution to the compound score (red).

The high top-5 accuracy and intersection support our analysis of Figure 3, demonstrating how testing the best recommendations has a high chance of providing optimal results. The bars are colored depending on the associated property weight, and thus, the associated contribution to the compound score (colored in red).

## 5 CONCLUSION

To conclude, we have introduced a novel approach called X-PCR, which – to the best of our knowledge – is the first explainable and resource-aware take on the task of model selection. While we here successfully applied it to the domain of time series forecasting with DNNs, our methodology can be easily generalized for other learning tasks. X-PCR recommends models with the help of a meta-learning paradigm that pays close attention to multiple objectives grouped into aspects of predictive error, complexity, and resources. Our solutions for calculating index values and compound scores enable improved comparability of properties and are beneficial additions to the existing work on assessing the energy efficiency of ML. Any model recommendation is accompanied by multi-level explanations and the interactiveness of our framework allows users to control and understand the model selection process. In our extensive evaluation, we provided an overview of the performance trade-offs taking place in time series forecasting and demonstrated the realizability of X-PCR selection. Considering model complexity and resource efficiency in addition to predictive precision has been shown to completely change the understanding of SOTA in forecasting. Our method outperformed competing approaches and was able to achieve near-to-optimal predictive performance (85%) while only requiring a fraction of the computational effort (20%). We deem our approach highly beneficial for the domain of meta-learning and model selection, as well as time series forecasting. For future work, we intend to apply X-PCR to even more data, their respective meta-features, and models. With our work, we hope to contribute to making both time series analysis and ML in general more resource-aware and trustworthy.

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
