# OpenReview forum: "Explainable Multi-Objective Model Selection for Time Series Forecasting"
_ICLR.cc/2024/Conference — ICLR 2024 Conference Withdrawn Submission_

### Official Review · Reviewer_Sacq · 2023-10-27

**Soundness:** 2 fair
**Presentation:** 2 fair
**Contribution:** 2 fair
**Rating:** 5
**Confidence:** 3

**Summary:**

This paper proposes a novel method, X-PCR, for explainable mulit-objective model selection and uses it for time series forecasting. The contribution comes from several folds, 1) it is the first explainable and resource-aware model selection framework, 2) successfully applies it to the task of time series forecasting, 3) adequate experiments -- 1000 experiments across 114 data sets.

**Strengths:**

The paper proposes a promising direction conducive to the democratization of machine learning methods.
It is novel to consider computing resources when designing the objective function for model selection.
Besides the time series forecasting, X-PRC can be easily generalized for other learning tasks.
The authors conducted experiments under a large amount of data and experimental conditions.

**Weaknesses:**

The details of the paper are not clear enough, such as how to describe the consumption of computing resources in PCR function.
Does different scales among multiple PCR functions cause problems for optimization?
I have some concern about the efficiency of this method due to a exhaustive search.
I don't particularly understand where the interpretability of the method is reflected. Can it give some evidenc or confidence interval of model selection results?

**Questions:**

The details of the paper are not clear enough, such as how to describe the consumption of computing resources in PCR function.
Does different scales among multiple PCR functions cause problems for optimization?
I have some concern about the efficiency of this method due to a exhaustive search.
I don't particularly understand where the interpretability of the method is reflected. Can it give some evidenc or confidence interval of model selection results?

---

> ### Author Response · Authors · 2023-11-16
>
> - resource consumption was profiled via CodeCarbon, as mentioned in Sec 4
> - scaling of PCR functions is taken care of by relative index scaling, as explained in Sec 3.2
> - exhaustive search is not efficient at all, hence we only used it as a baseline approach for model selection
> - we explained why we understand our method as interpretable in Sec 3.4, and tried to exemplary show this practically in Fig 5
>
> It is very frustrating to see copy-paste reviews like this (weaknesses & questions)

---

### Official Review · Reviewer_m8sj · 2023-10-29

**Soundness:** 1 poor
**Presentation:** 2 fair
**Contribution:** 1 poor
**Rating:** 3
**Confidence:** 4

**Summary:**

This paper proposed a method to estimate forecasting model’s performance, complexity (number of parameters and model size)and resource consumption (training and inference time, power draw) based on dataset meta-features. Then the predicted numbers enable multi-objective selection based on a weighting of different objectives. The authors tested their method on Monash datasets and it beats competing approaches such as AutoML and achieves 85% of predictive performance at only 20% of the computation cost required for exhaustive search.

**Strengths:**

The resource aware perspective is interesting and relevant. The provided code repo and demo greatly improves reproducibility.

**Weaknesses:**

The idea of using dataset meta-features to find good models for a new dataset is not new. It’s been explored in many transfer HPO methods. The earliest work to me is [1] where the meta-features are used to find a good initialization HP. Many methods not based on meta-features are also proposed afterwards such as [2] (There is a long list in this field, just to give some examples on how early this idea has been explored). The difference to this work is 1) most of them do not focus on time series dataset and 2) they work on learning good hyperparameters instead of model choice.

Following the above, the model choice is just the first step and the hyperparameters of the models play an important role in the final performance and resource consumption.

[1] M. Feurer, J. T. Springenberg, and F. Hutter, “Using meta-learning to initialize bayesian optimization of hyperparameters,” in ECAI workshop on Metalearning and Algorithm Selection (MetaSel), 2014, pp. 3–10

[2] Wistuba, Martin, Nicolas Schilling, and Lars Schmidt-Thieme. "Learning hyperparameter optimization initializations." 2015 IEEE international conference on data science and advanced analytics (DSAA). IEEE, 2015.

The other concern is the unnecessary complexity. For a fixed model, should the per-sample inference time and power draw be more or less the same? For those models in GluonTS, they have fixed context and prediction length and looking at the per-sample metric would make the usage of the predictor unnecessary. Also, why does the number of parameters and model size need to be predicted? They should be the same for all datasets. For the training time, why not also look at per-sample time? They should also be similar across the dataset. Finally, looking at Figure 6, predicting a model's performance based on the proposed dataset features seems the hardest and that’s where we need to be the most accurate.

In the end, I think it makes more sense to compare to transfer or meta-learning AutoML methods in this setting. It’s not a fair comparison in the current setting for AutoML.

**Questions:**

PCR is only mentioned firstly at Section 3.1 and should be explained much earlier.

How do the authors come up with the weights in Table 1?

The resource consumption is evaluated on one hardware, how much of the conclusion can be generalized to other hardwares? Will the order of the models change on a different hardware?

The model is currently encoded as a one-hot vector, but the number layers, parameters etc. should be part of the input features rather than the property to be predicted.

---

> ### Author Response · Authors · 2023-11-16
>
> ### Regarding identified weaknesses:
>
> - We are thankful for the proposed related literature we missed out on, however - from a first look - they suffer from the same issue as the works discussed in Sec 2 - overly focusing on predictive capabilities.
>
> - Per-sample inference time and power draw are strongly correlated, but not necessarily the same, since different models might utilize the hardware more or less efficiently.
>
> - It is indeed true that predicting a model's predictive performance is hardest (as shown in Fig 6), however the point we want to make is that model selection and meta-learning should still consider resource trade-offs and efficiency.
>
> ### Regarding questions:
>
> - The weights were selected to establish a good trade-off among all properties.
> - The other feedback shall be adressed in future work.

---

### Official Review · Reviewer_4DT3 · 2023-11-01

**Soundness:** 2 fair
**Presentation:** 2 fair
**Contribution:** 1 poor
**Rating:** 3
**Confidence:** 3

**Summary:**

This paper proposes an explainable model selection method, X-PCR to provide understandable and controllable recommendations of DNNs on time series forecasting tasks. X-PCR uses meta-learning to assess the suitability of any DNN in terms of (p)redictive error, (c)omplexity and (r)esource demand. X-PCR is tested on 114 data sets considering 11 DNNs. The experiment show X-PCR outperforms the random selection strategy and AutoKeras [1]

[1]. Haifeng Jin, Qingquan Song, and Xia Hu. Auto-keras: An efficient neural architecture search system. In Proceedings of the 25th ACM SIGKDD international conference on knowledge discovery & data mining, pp. 1946–1956, 2019.

**Strengths:**

The topic is interesting. The idea of designing a user-centric XAI framework is novel to some extent.

**Weaknesses:**

I have lost the novelty of this study. The current claim that "X-PCR is the first explainable and resource-aware model selection" is not convincing as I didn't see a clear motivation for designing an explainable and resource-aware model selection method.

The proposed idea lacks support from my perspective. For example, how the authors determine to choose the three aspects of (p)redictive error, (c)omplexity and (r)esource demand. Are they from a survey from practice or from the existing literature review?

As X-PCR is a model selection method, it is compared to a random model selection strategy but not another model selection strategy. Therefore, it is unclear how efficient X-PCR on model selection.

**Questions:**

Why use AutoKeras as the baseline method? If you think the NAS system can be the baseline why not use the SOTA method that is specifically designed for time series such as [2]?
Why not compare X-PCR to other model selection methods?

[2] Lyu, Zimeng, and Travis Desell. "ONE-NAS: an online neuroevolution based neural architecture search for time series forecasting." Proceedings of the Genetic and Evolutionary Computation Conference Companion. 2022.

---

> ### Author Response · Authors · 2023-11-16
>
> ### Comments regarding identified weaknesses:
>
> - "clear motivation for designing an explainable and resource-aware model selection method" was given in Sec 2.3, based on extensive related work
>
> - the three focused aspects were proposed based on personal experience and related work
>
>
> ### Regarding questions:
>
> - AutoKeras was chosen because autoML is another approach to model selection
> - It is not feasible to test and compare ourselves with the suggested method [2], as it does not come with public implementation